# Biosynthesis of Gamma-Aminobutyric Acid (GABA) by *Lactiplantibacillus plantarum* in Fermented Food Production

**Massimo Iorizzo** , **Gianluca Paventi** * and **Catello Di Martino**

Department of Agricultural, Environmental and Food Sciences, University of Molise, Via De Sanctis, 86100 Campobasso, Italy; iorizzo@unimol.it (M.I.); lello.dimartino@unimol.it (C.D.M.)
* Correspondence: paventi@unimol.it

**Abstract:** In recent decades, given the important role of gamma-aminobutyric acid (GABA) in human health, scientists have paid great attention to the enrichment of this chemical compound in food using various methods, including microbial fermentation. Moreover, GABA or GABA-rich products have been successfully commercialized as food additives or functional dietary supplements. Several microorganisms can produce GABA, including bacteria, fungi, and yeasts. Among GABA-producing microorganisms, lactic acid bacteria (LAB) are commonly used in the production of many fermented foods. *Lactiplantibacillus plantarum* (formerly *Lactobacillus plantarum*) is a LAB species that has a long history of natural occurrence and safe use in a wide variety of fermented foods and beverages. Within this species, some strains possess not only good pro-technological properties but also the ability to produce various bioactive compounds, including GABA. The present review aims, after a preliminary excursus on the function and biosynthesis of GABA, to provide an overview of the current uses of microorganisms and, in particular, of *L. plantarum* in the production of GABA, with a detailed focus on fermented foods. The results of the studies reported in this review highlight that the selection of new probiotic strains of *L. plantarum* with the ability to synthesize GABA may offer concrete opportunities for the design of new functional foods.

**Keywords:** *Lactobacillus plantarum*; functional food; L-glutamate decarboxylase; lactic acid bacteria

## 1. Introduction

Gamma ($\gamma$)-Aminobutyric acid (GABA), also named 4-aminobutyric acid, is a four-carbon non-protein amino acid that is widely distributed in an extensive variety of organisms including algae, bacteria, fungi, animals, plants, and cyanobacteria [1–7].

Although GABA is present in many foods such as fruits, vegetables and grains, its content in them is relatively low [8,9]. As a result, over the years many studies have been devoted to the most suitable strategies to increase the amount of GABA in food [10,11] such as through chemical synthesis [12], plant enrichment [13], or microbial fermentation [3].

Microbial synthesis of GABA may be much more promising than chemical synthesis methods since the former is characterized by high specificity, environmental friendliness and cost-effectiveness [3].

In addition, GABA production by beneficial and pro-technological microorganisms has the potential to increase the functional effect of some fermented foods and beverages [11,14]. So far, various studies have confirmed that several microorganisms like fungi, bacteria, and yeasts have the ability to synthesize GABA [3,15,16].

Lactic acid bacteria (LAB) are ubiquitous microorganisms and are often naturally present in some traditional fermented foods as well. Many LAB species are used as starters in some industrial food fermentations for their pro-technological properties [17,18].

Some LAB species are capable of producing high amounts of GABA [19–21] and could be exploited for the production of GABA-fortified foods [14].

Among GABA-producing LAB, *Lactiplantibacillus plantarum* (formerly *Lactobacillus plantarum*) is a facultative heterofermentative species with high adaptability to many different conditions, being isolated from various ecological niches including milk, fruit, cereal crops, vegetables, bee bread, fresh meat [22–25] and fermented foods [26,27]. This bacterial species *L. plantarum* is a normal inhabitant of the gastro-intestinal tract of insects, fish and mammals, including humans [28–32] and is included in the QPS (Qualified Presumption of Safety) and in GRAS (Generally Recognised as Safe) lists [33,34].

Because of many of its intrinsic properties, numerous strains belonging to this species are proposed as animal and human probiotics [31,32,35–40].

*L. plantarum* is widely used as a starter culture in the fermentation of raw materials from plant and animal origin, where it contributes to enhancing the sensorial quality and shelf life of fermented products [38,39,41–44]. Some *L. plantarum* strains also increase the functional properties of various fermented foods by producing a variety of bioactive compounds, including GABA [19,45].

The present review aims, after a preliminary excursus on the function and biosynthesis of GABA, to provide an overview of the current uses of microorganisms and, in particular, of *L. plantarum* in the production of GABA, with a detailed focus on fermented foods.

## 2. GABA Function and Metabolism

GABA is produced by bacteria [3,46] fungi [47,48], plants [49,50], vertebrate animals and invertebrates [51–53]. Furthermore, Archaea possesses enzyme genes involved in GABA biosynthesis [54–56].

Due to this pervasive presence in biological kingdoms and ecosystems, we tend to consider the GABA molecule more as a ubiquitous signaling molecule than as a specific synaptic neurotransmitter [57,58].

GABA-mediated interregnum communication has been observed between algae and invertebrates [59], plants and fungi [60], plants and insects [61], and plants and bacteria [62].

In plants, GABA is an endogenous signaling molecule involved in various physiological and biochemical processes that promote plant growth and development, and mediate responses to abiotic and biotic stresses, including pathogen and insect attacks [1,63,64]. In addition, GABA improves photosynthetic processes, inhibiting the production of reactive oxygen species (ROS), activating antioxidant enzymes, and regulating stomatal opening in case of water stress [65].

In plants, GABA is synthesized from glutamate or arginine and transferred by GABA-permease to mitochondria, where GABA is catabolized by GABA transaminase and succinate semialdehyde dehydrogenase to succinate. The succinate enters the tricarboxylic acid (TCA) cycle to maintain the C/N balance in cells [1].

Over the last several decades GABA has attracted great attention due to its many positive effects on mammalian physiology [10,15,66].

As known, in fact, GABA is the most common inhibitory neurotransmitter in the human central nervous system [67]. Furthermore, besides being an important antidepressant [68], GABA also performs other functions including neuroprotective, anti-inflammatory, antioxidant and antihypertensive effects [66], enhancement of immunity under stress conditions [69], prevention of cancer cell proliferation [70], prevention of diabetic conditions [71], and cholesterol-lowering effect [72].

In mammalian, GABA is synthesized from L-glutamate in the cytoplasm of neuronal and glial cells by the enzyme glutamate decarboxylase (GAD; EC4.1.1.15) using pyridoxal 5′-phosphate (PLP) as an enzyme cofactor [73] (Figure 1). GABA can also be synthesized through deamination and decarboxylation reactions of putrescine, spermine, spermidine, ornithine, and L-glutamine [2].

As for mammalian species, in microorganisms, GABA is produced from L-glutamate through a GAD enzyme-mediated decarboxylation [58] with PLP as a cofactor (Figure 1).

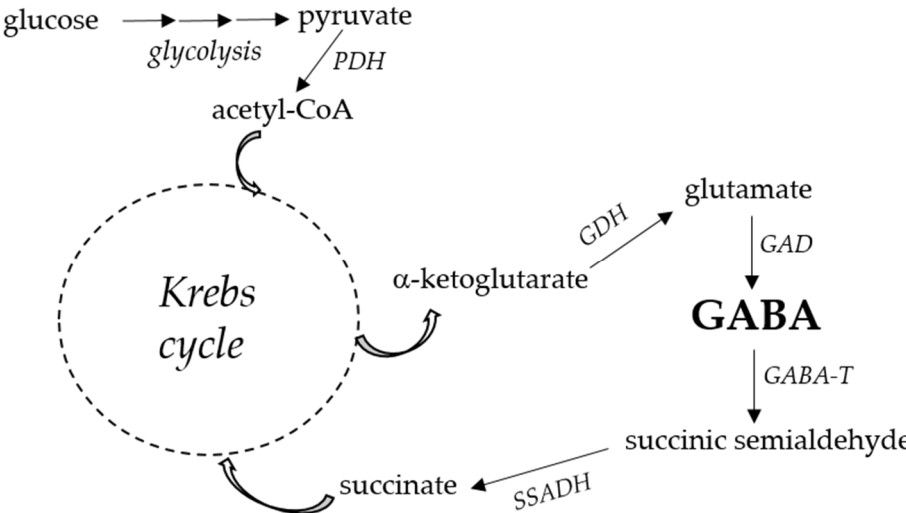

**Figure 1.** GABA production from L-glutamate by glutamate decarboxylase (GAD) with pyridoxal-5′-phosphate (PLP) as a cofactor.

Through a well-established pathway of enzymes known as GABA shunt (Figure 2), some bacteria catabolize GABA [74,75]. The GABA shunt is characterized by a group of enzymes that convert GABA to succinate to fuel the tricarboxylic acid (TCA) cycle in the production of energy and essential metabolic intermediates as carbon skeletons for the cell. In the shunt of GABA, microbial enzymes such as GABA transaminase and succinic semialdehyde dehydrogenase have an optimal pH in the alkaline range of around 8. These observations have led some scholars to advance the hypothesis that the GABA shunt, in addition to representing the link between nitrogen and carbon metabolism, has an important function in the maintenance of pH homeostasis in acidic environments [75,76].

**Figure 2.** Metabolic pathway of GABA production from the TCA cycle (adapted from Sahab et al. [11]). For higher clarity, this scheme reports only enzymes and relevant substrates/products, omitting coenzymes and other compounds involved in the reactions showed. Abbreviations: PDH, pyruvate dehydrogenase; GDH, glutamate dehydrogenase; GAD, glutamate decarboxylase; GABA-T, GABA transaminase; SSADH, succinic semialdehyde dehydrogenase.

As reported above, microbial GABA synthesis is strictly dependent on the GAD enzyme which is encoded by *gadA* or *gadB* genes in bacterial cells. Glutamate is transported into a cell through an antiporter, and then decarboxylation occurs. Finally, the GABA product is secreted from the cell by the glutamate/GABA antiporter, which is encoded by the *gadC* gene [77].

In recent years, many researchers have studied *L. plantarum* for its ability to synthesize GABA using the GAD system.

Only *Levilactobacillus brevis* possesses two GAD genes that produce isozyme GADs among the LAB examined so far [78,79]. The glutamic acid decarboxylase (GAD) system encoded by the gad operon is responsible for glutamate decarboxylation and GABA secretion in bacteria and consists of two important elements: Glu/GABA antiporter *gadC* and the glutamate decarboxylase enzyme encoded either by *gadA*, *gadB* genes [80]. This system

converts glutamate into GABA and while doing so consumes protons thus maintaining cytosolic pH homeostasis [79].

Unlike *L. brevis*, *L. plantarum* has only one GAD coding enzyme in its genome, the *gadB* and there may not be a specific glutamate/GABA antiporter (*gadC*) gene [80].

In a study conducted by Nakatani et al. on the genome of *L. plantarum* KB1253, it was found that this strain contains two *gadB* genes coding for glutamate decarboxylase [81].

Many studies showed that *L. plantarum* can produce appreciable amounts of GABA, so there must be a transporter responsible for transporting glutamate and GABA in and out of the cell. A glutamate/gamma-aminobutyrate transporter family protein coded by the *yjeM* gene can be the best candidate for such a transporter [82].

Further investigation, conducted by Surachat et al. indicates that *L. plantarum* is a key GABA-producing species in nature since almost all strains encoded the GAD operon in their genome [83].

## 3. Production of GABA by Microorganisms

GABA can be obtained not only from natural sources but also through plant enrichment, chemical synthesis, enzymatic process and microbial metabolism [15,84]. Due to the low GABA content in natural animal- and plant- associated food products, high GABA-producing microorganisms are of great importance to produce food-grade GABA and GABA-rich fermented foods via fermentations [85].

The biosynthesis of GABA by microorganisms is safe and eco-friendly and provides the possibility of production of new naturally fermented health-oriented products [16,86].

### 3.1. Production of GABA by Fungi

Other than bacteria, various yeasts and molds that belong to the kingdom of fungi, have also been reported to as able to produce GABA.

Some *Rhizopus oligosporus* and *Rhizopus oryzae* strains have been shown to produce GABA during tempeh fermentation (fermented soybean) [87].

Similarly, *Rhizopus monosporus* strain 5351 has been reported to increase GABA content in soybean and mung bean [88,89].

Marine yeasts *Pichia guilliermondii* and *Pichia anomala* isolated from the Pacific Ocean off Japan have high GABA-producing abilities [90,91].

*Actinomucor elegans* AS 3.227 has been reported to increase the GABA concentration in sufu (traditional fermented soybean food from China) manufacturing using solid-state fermentation [92].

Glutamic acid decarboxylase has also been identified in yeasts such as *Saccharomyces cerevisiae* and *Kluyveromyces marxianus* isolated from fermented products [93–95].

Other yeasts belonging to the species *Kazachstania unispora*, *Sporobolomyces carnicolor*, *Sporobolomyces ruberrimus*, *Nakazawaea holstiiand*, and *Pichia scolyti*, isolated from wild flowers, also have GAD activity [96].

*Aspergillus oryzae* NSK is a GABA-generating mold used as a starter culture to ferment rice koji for sake production and soy sauce koji [97–101].

Cai et al. demonstrated that oats fermented by *A. oryzae* var. *effuses* 3.2825, *A. oryzae* 3.5232 and *R. oryzae* 3.2751 can be recommended as tempeh-like functional foods with higher GABA [102].

In other studies, fermentations by *Monascus pilosus* IFO 4520 [103] and *Monascus purpureus* CCRC 31615 [104] increase the content of γ-aminobutyric acid (GABA) in the beni-koji and in the fermented rice.

### 3.2. Production of GABA by Bacteria

GABA is naturally synthesized by several bacteria. Indeed, not all strains within one species can produce GABA, as the ability depends on the presence of GAD genes and glutamate/GABA antiporter [74].

*Bacillus* is a commonly reported bacteria that can produce GABA [105,106]. Besides *Bacillus*, *Corynebacterium glutamicum* was found to produce endogenous L-glutamate [107], *Streptomyces bacillaris* and *Streptomyces cinereus* were reported to increase the GABA content in fermented tea [108]. Otaru et al. have shown that human intestinal *Bacteroides* are able to synthesize GABA [109].

Recent studies revealed that the increased level of GABA in the human gut could be derived from the ability of the intestinal microbiota or ingested probiotics, such as *Bacteroides*, *bifidobacteria*, and some LAB (*lactobacilli*), to metabolize dietary monosodium glutamate [109–111]. Therefore, numerous research has been directed towards isolating and characterizing GABA-producing bacteria to be used as starters for the production of GABA-enriched fermented food [3].

Because of their GRAS status, some LAB are widely used in the production of fermented foods [18] and act as potential probiotic cultures. Actually, in addition to protechnological functions, LAB also offer beneficial functions such as antioxidant and antimicrobial activities, as well as the formation of bioactive compounds such as GABA [112,113].

Therefore, the use of GABA-producing LAB has been considered a promising possibility in order to increase the nutritional, functional, sensory and technological properties of some fermented food products [10,19,114].

GABA can be biosynthesized by various LAB strains mainly belonging to the genera of *Lactobacillus*, *Lactococcus*, *Pediococcus*, *Leuconostoc*, *Enterococcus*, *Streptococcus*, *Weissella*, *Lacticaseibacillus*, *Lactiplantibacillus* and recently, *Levilactobacillus* and *Secundilactobacillus* [21,115–121].

Nowadays, *Lactiplantibacillus plantarum* (formerly classified as *Lactobacillus plantarum*) is among the main LAB species proposed to be used as probiotic starter cultures to produce GABA in the fermented food and beverage industry [35,122,123].

## 4. Production of GABA by *L. plantarum*

The production of GABA varies among various LAB strains and is affected by several factors such as pH, fermentation temperature, fermentation time, L-glutamic acid concentration, media additives, and carbon and nitrogen sources [3,85,114,124]. The optimization of these parameters could maximize the amount of GABA contained in some LAB-fermented foods [3,112].

In recent years, many researchers have studied *L. plantarum*, in particular, for its ability to synthesize GABA in different substrates and growing conditions.

Table 1 summarizes the results of studies investigating the ability of different strains of *L. plantarum* to produce GABA in different growing media.

**Table 1.** GABA production by *Lactiplantibacillus plantarum* (previously *Lactobacillus plantarum*) in different culture media.

| Microorganism | Isolation Source | Culture Medium | GABA Production | Comments | Refs. |
|---|---|---|---|---|---|
| *L. plantarum* C48 | cheese | MRS | 16.0 mg/kg | Survival and GABA production in simulated GI conditions | [115] |
| *L. plantarum* CCARM 0067 | CCARM | CDM | ≈700 mM (48 h) | Anti-proliferative and anti-metastatic activity in HT-29/5FUR cell line | [70] |
| *L. plantarum* DM5 | Marcha of Sikkim | MRS + 100 mM MSG | not quantified | GABA production has been qualitatively identified by the TLC | [125] |
| *L. plantarum* KCTC 3103 | Unknown | MRS modified | 0.67 g/L | Two-stage fermentation: cell grown (stage 1); GABA production (stage 2) | [126] |
| *L. plantarum* K154 | kimchi | broth fortified with skim milk and 2% MSG | 15.53 mg/mL | Co-culture with *Ceriporia lacerata* | [127] |
| *L. plantarum* EJ2014 | Rice bran | SM | 19.8 g/L | Optimization of production by the addition of yeast extract | [124] |

**Table 1.** *Cont.*

| Microorganism | Isolation Source | Culture Medium | GABA Production | Comments | Refs. |
|---|---|---|---|---|---|
| *L. plantarum* K154 | kimchi | MRS + 30 g/L MSG | 0.2 g/L | Potential probiotic: good resistance to vancomycin and polymyxin B, tolerance to bile juice and low pH | [128] |
| *L. plantarum* Taj-Apis362 | honeycomb and stomach of honeybee | MRS + 50 mM MSG | 7.15 mM | culture temperature of 36 °C, initial pH of 5.31 and incubation time of 60 h | [129] |
| *L. plantarum* 45a | cambodian fermented foods | MRS + 2% MSG | 20.34 mM | Two other strains of *L. plantarum* capable of synthesizing GABA have been identified: 44d (16.47 mM GABA) and 37e (5.63 mM GABA) | [130] |
| *L. plantarum* FNCC 260 | indonesian fermented foods | MRS + 25–100 mM MSG | 809.2 mg/L | MSG, PLP, and pyridoxine were shown to positively affect GABA production | [131] |
| *L. plantarum* BC114 | Sichuan paocai (fermented vegetable) | MRS + 20 g/L MSG | 3.45 g/L | *L. plantarum* BC114 highlighted the ability to produce GABA and reduce nitrates | [132] |
| *L. plantarum* LSI2-1 | Thailand fermented food | GYP + 3% MSG | 22.94 g/L | Only the gadA as glutamate decarboxylase (GAD) was found in the genome | [133] |
| *L. plantarum* MNZ | fermented soybean | MRS | 3.96 mM | 6% glucose, 0.7% ammonium nitrate, pH 4.5 and temperature 37 °C. | [134] |
| *L. plantarum* K255 | kimchi | MRS + 3% MSG | 821.2 µg/mL | the K255 strain was incubated at 37 °C for 18 h. | [135] |
| *L. plantarum* FBT215 | kimchi | MRS modified (1% fructose; 2% tryptone, 50 mM MSG) | 103.7 µg/mL | PLP is a major factor influencing GABA production | [123] |
| *L. plantarum* B-134 | Makgeolli | MRS + 3% MSG | 25 mM | optimum culture condition: 37 °C, pH 5.7 without NaCl | [136] |
| *L. plantarum* N1-2 | Nham | MRS + 5% MSG | 0.13 mg/10 g | pH of 5.7, without NaCl | [137] |
| *L. plantarum* Y7 | kimchi | MRS modified (2% fructose, 2% peptone and 175 mM MSG) | 4.9 µg/mL | culture conditions: 37 °C, pH 6.5, and 48 h. | [138] |
| *L. plantarum* L10-11 | Plaa-som | MRS + 4% MSG | 15.74 g/L | addition of NaCl by up to 7% (*w/v*) did not suppress GABA production | [139] |
| *L. plantarum* FRT7 | Paocai | MRS 3% MSG and 2 mmol/L of PLP | 1158.6 mg/L | 40 °C; pH of 7.0 for 48 h | [140] |
| *L. plantarum* HUC2W | | MRS + 4% MSG | 3.92 g/L | at 37 °C for 24 h | [141] |

Abbreviations: MRS, de Man, Rogosa and Sharp medium; GI, gastrointestinal; CCARM, Culture Collection of Antimicrobial Resistant Microbes; CDM, chemical defined medium; HT-29/5FUR, human colon adenocarcinoma cell line (HT-29) resistant to 5-fluorouracil (5-FU); MSG, mono-sodic glutamate; TLC, thin layer chromatography; SM, synthetic medium (consisting of 100 g/L Yeast extract, 10 g/L dextrose, and 22.5 g/L MSG); PLP, pyridoxal 5′-phosphate; GYP, Glucose-yeast extract-peptone; GAD, glutamic decarboxylase.

The most commonly used culture medium is MRS (de Man, Rogosa and Sharp), a standard substrate designed to promote LAB growth [142]. Monosodium glutamate (MSG), as a source of L-glutamine, is usually supplemented directly into MRS to enhance GABA synthesis from *L. plantarum* strains [82].

However, the optimal concentration of MSG depends on the bacterial strain. For example, Yogeswara et al. investigated the GABA production from *L. plantarum* FNCC 260 strain using a wide range of MSG concentrations. The results showed a maximum GABA production (1226 mg/L) by adding 100 mM of MSG to the MRS medium and then incubating at 37 °C for 108 h [131].

In another study, after 18 h at 34 °C, *L. plantarum* K74 produced 134.52 µg/mL of GABA in MRS broth containing 1% MSG, 212.27 µg/mL of GABA in MRS broth containing 2% MSG, and 234.63 µg/mL of GABA in MRS broth containing 3% MSG [135].

Gomaa et al. examined the effect of MSG and PLP on GABA production from *L. brevis* and *L. plantarum* strains, isolated from Egyptian dairy products. The culture medium

used was the following composition: 50 g/L glucose; 25 g/L soya peptone; 0.01 g/L MnSO$_4$C$_4$H$_2$O and 2 mL Tween 80. The results of the aforementioned study show that the amount of extracellular GABA produced is proportional to the amounts of MSG and PLP added. Co-culture of *L. brevis* and *L. plantarum* produced the highest amount of GABA, 160.57 mM and 224.69 mM, in the presence of 750 M MSG and 200 μM PLP, respectively [143].

Park et al. have obtained high amounts of GABA (19.8 g/L) at 30 °C from *L. plantarum* EJ2014 using the following culture medium: 100 g/L Yeast extract, 10 g/L dextrose, and 22.5 g/L (*w/v*) MSG [124].

In a study conducted by Shan et al. *L. plantarum* NDC75017 produced 3.2 g/kg of GABA, at 30 °C for 48 h, in skimmed milk with 80 mM MSG and 18 μM PLP [144].

As evidenced in all the studies mentioned above, the amount of monosodium glutamate initially available is an important factor in the production of GABA [145].

In fact, as also confirmed in other studies cited below, an initial excessive concentration of MSG may inhibit cell growth or inhibit GABA production due to osmotic stress, while a low concentration of MSG may not meet the requirements of high GABA production [146]. As far as the incubation time is concerned, we have observed that the amount of GABA after reaching the maximum amount after a certain period of time, tends to decrease subsequently. This effect may be caused by a lower availability of precursors (e.g., MSG) but also be linked to degradation, by GABA aminotransferase, of GABA to succinic semialdehyde, which is subsequently converted by succinic semialdehyde dehydrogenase for entry into TCA [11].

Temperature and pH have been reported as the main environmental factors that can modulate gad gene expression [147]. Therefore, adjusting pH and temperature during fermentation is a very effective way to increase microbial GABA production.

LAB employ a complex but efficient combination of different acid resistance systems [148].

Among the various types of tolerance mechanisms to the acidic environment, the GAD system is considered one of the most effective acid mitigation pathways.

In this system, intracellular protons are consumed through decarboxylation of glutamate in the cytoplasm [74].

Shin et al. showed that 40 °C and a pH of 4.5 were the best parameters for the expression of *gadB* gene encoding GAD from *L. plantarum* ATCC 14,917 in *E. coli* BL21 (DE3) [149].

Variation in pH enhances activation of the GAD pathway since it is considered one of the mechanisms that preserve cell homeostasis [150]. Wu et al. evaluated the performance of the GAD pathway in comparison with other acid resistance mechanisms and highlighted how the GAD system is an essential mechanism to maintain metabolic activity under intra- and extracellular acidity [79].

Therefore, the pH of the environment is crucial for the synthesis of GABA. However, it seems that this depends on the bacterial strain [149].

Zhang et al. tested how initial pH affects GABA production by *L. plantarum* BC114. The best concentration of GABA was detected at pH 5.5, obtaining double the amount of GABA yielded at pH 4.0 [132]. Similar results have been obtained in other studies [129,139,140].

Tajabadi et al. found that after 60 h *L. plantarum* Taj-Apis362 produces the highest amount of GABA (7.15 mM; 0.74 g/L) at 36 °C in modified MRS: 497.97 mM glutamate, pH 5.31 [129]. Tanamool et al. found that the highest GABA production (15.74 g/L) by *L. plantarum* L10-11 cultured in MRS with 4% MSG at 30 °C was obtained within 48 h, with a pH range of 5–6 [139].

Very recently, Cai et al. reported that *L. plantarum* FRT7 after 48 h produced approximately 1.2 g/L in MRS supplemented with 3% MSG and 2 mmol/L of PLP at 40° C with an initial pH of 7.0 [140].

In a recent study conducted by Kim J et al., the optimal conditions for efficient GABA production by *L. plantarum* FBT215 in modified MRS broth containing 50 mM

MSG were investigated. Therefore, the optimal culture temperature for GABA production (103.67 μg/mL) was 37 °C and this efficiency was highest at pH 7.5 and 8.5 and decreased under acidic conditions [123].

Instead, Yogeswara et al. found that GABA production from *L. plantarum* FNCC 260 was greatly improved under acidic conditions (pH 3.8) in Pigeon pea (*Cajanus cajan*) milk fermentation [151]. This result is in line with a previous study by Yogeswara et al. where maximum GABA production from *L. plantarum* FNCC 260 in MRS was observed at pH 4.0 [131].

Regarding the temperature, Yang et al. reported that GAD functionality is directly related to an increase in temperature until it reaches an optimum, after which GAD activity decreases until thermal inactivation [152]. Another study with *L. plantarum* showed an increase in GAD activity up to 40 °C, achieving optimal GABA production at 35 °C [144].

Importantly, *L. plantarum* is a mesophilic bacterium with an optimal growth temperature of around 37 °C. This evidence explains why, in all the studies cited in this review, the optimal temperatures for maximum GABA production were in the range of 30–40 °C.

*GABA Production by L. plantarum in Fermented Foods*

According to the available data, naturally occurring GABA in foods is usually low [85,153]; therefore, the food industry has shown great interest in GABA-enriched foods, through microbial fermentation.

Currently, *L. plantarum* is a LAB species commonly found in various fermented foods and beverages. Therefore, some food scientists have proposed strains of *L. plantarum* as starters in single culture (Table 2) or in co-culture with other microbial species (Table 3) to enrich GABA in some traditional or innovative fermented foods, particularly from plant-based sources.

Table 2 summarizes the results obtained from the use of *L. plantarum* as a single starter in different fermented foods.

**Table 2.** GABA production by *Lactiplantibacillus plantarum* (previously *Lactobacillus plantarum*) in different fermented foods.

| Microorganism | Isolation Source | Fermented Food | GABA Production | Comments | Refs. |
|---|---|---|---|---|---|
| *L. plantarum* C48 | cheese | buckwheat, amaranth, chickpea and quinoa flours | 504 mg/kg in bread | Good organoleptic properties of bread enriched of GABA | [154] |
| *L. plantarum* DSM19463 | cheese | grape must | 8.9 g/kg in fermented grape must | In vitro potential anti-hypertensive effect and dermatological protection. | [155] |
| *L. plantarum* KB1253 | pickles | tomato juice | 41 mM | GABA-enriched fermented tomato juice | [156] |
| *L. plantarum* KCTC 3105 | Unknown | soya milk | 424.67 μg/g DW | Soya yogurt with high levels of GABA, produced using a co-culture of *L. acidophilus*, *L. plantarum* and *L. brevis* strains | [157] |
| *L. plantarum* NDC75017 | fermented milk | 12% skim milk + 80 mM MSG | 314.56 mg/100 g | Good flavor and texture of fermented milk-based product | [144] |
| *L. plantarum* NTU102 | cabbage pickles | 8% skim milk + 1% (*w/v*) MSG | 629 mg/L | together with GABA, production of ACEI was also found, suggesting a possible use of fermented products as potential functional food (hypertension regulation) | [158] |
| *L. plantarum* C48 | cheese | wholemeal wheat flour | 100 mg/K | low ACE inhibitory activity (15%) due to synthesis of ACEI | [159] |
| *L. plantarum* GB01-21 | | cassava powder | 80.5 g/L 2.68 g/L h (productivity) | two-step production with *Corynebacterium glutamicum* G01 (to produce glutamate) and *L. plantarum* GB01-21 | [152] |

**Table 2.** *Cont.*

| Microorganism | Isolation Source | Fermented Food | GABA Production | Comments | Refs. |
|---|---|---|---|---|---|
| *L. plantarum* Dad-13 | FNCC | pigeon pea milk | 5.6 g/L | The supplementation of sucrose, MSG, and whey isolate significantly increased GABA levels in fermented pigeon pea | [151] |
| *L. plantarum* NRRL B-59151 | | FOE and HFOE (oat) | GABA content: 7.35 mg/100 g in FOE and 8.49 mg/100 g in HFOE | Fermented oat demonstrated antidiabetic effects | [160,161] |
| *Lactobacillus plantarum* HU-C2W | | litchi juice | 134 mg/100 mL | Fermentation condition: 37 °C for 40 h | [141] |
| *L. plantarum* DW12 | fermented red seaweed | red seaweed + 1% MSG | 4 g/L | Fermentation at 30 °C after 60 days. Substrate composition: red seaweed, cane sugar and potable water in a ratio of 3:1:10, pH 6 | [162] |
| *L. plantarum* DW12 | fermented red seaweed | red seaweed + 0.5% MSG | 1284 mg/L | Fermentation at 30 °C after 60 days. Substrate composition: red seaweed, cane sugar and potable water in a ratio of 3:1:10, pH 6 | [163] |
| *L. plantarum* DW12 | fermented red seaweed | MCW + 0.5% MSG | 12.8 mg/100 mL | MCW supplemented with 0.5% MSG and 1% sugarcane, pH 6 after 72 h of fermentation | [164] |

Abbreviations: DW, dry weight; MSG, mono-sodic glutamate; ACEI, angiotensin converting enzyme inhibitor; ACE, angiotensin converting enzyme; FNCC, Food and Nutrition Culture Collection; HFOE, fermented oat + honey; FOE, Fermented Oat; MCW, mature coconut water.

In a recent study [151], it has been proposed a drink prepared from germinated pigeon pea (*Cajanus cajan*) and fermented using probiotic *L. plantarum* Dad-13, isolated from dadih, fermented buffalo milk [165]. *C. cajan* commonly known as pigeon pea, red gram or gungo pea is an important grain legume crop, particularly in rain-fed agricultural regions in the semi-arid tropics, including Asia, Africa and the Caribbean [166].

Additional nutrients such as MSG 1%, whey 4%, and sucrose 3% were added to pigeon pea extract and fermentation was carried out in a closed container at 30 °C for 48 h without shaking. Maximum GABA production (5.6 g/L) was obtained after 12 h of fermentation.

Wang et al. have shown that it is possible to increase the production of GABA in fermented lychee juice by *L. plantarum* HU-C2W [141]. Litchi (*Litchi chinensis* Sonn.) is a well-known tropical fruit originating from Asia [167]. After 40 h at 37 °C, a GABA content of 134 mg/100 mL was observed [141].

In various studies, *L. plantarum* DW12, isolated by Ratanaburee et al. from a fermented red seaweed, has been successfully used as probiotic and starter culture to produce fermented foods and beverages due to its safety aspects and ability to produce GABA [83,162–164].

The results obtained in [162] reported that *L. plantarum* DW12 produces 4 g/L GABA in red seaweed fermentation (red seaweed-cane sugar-potable water = 3:1:10, *w/w/v*) at 30 °C after 60 days. The red seaweed *Gracilaria fisheri* is commonly found along the coast of south-east Asian countries and used as a fresh vegetable and as a dried product [168].

In another study conducted by Hayisama-Ae et al., a novel functional beverage was produced from red seaweed *Gracilaria fisheri* (known as Pom Nang seaweed in Thailand), using *L. plantarum* DW12 as a starter culture [163]. Fermented red seaweed beverage was produced as follows: red seaweed, cane sugar and potable water in a ratio of 3:1:10 with an addition of 0.5% of MSG and an initial pH of 6.0. After 60 days the fermented red seaweed beverage (FSB) contained 1.28 g/L GABA.

A study conducted by Kantachote et al. aimed to add value to mature coconut water by using the probiotic *L. plantarum* DW12 for the production of GABA-enriched fermented beverages. Coconut water, with an initial pH of 5.0, was supplemented with

0.5% monosodium glutamate and 1% sugarcane and fermented from *L. plantarum* DW12. After 48 h, the fermented product contained 128 μg/mL of GABA [164].

Coconut (*Cocos nucifera* L.) is an important fruit tree found in tropical regions and its fruit can be made into a variety of foods and beverages [169].

Zarei et al. investigated the potential of GABA production by a *L. plantarum* strain in whey protein beverage [170], building on previous research, in which this strain, isolated from traditional doogh (yogurt, herbs and water) from west region of Iran, have shown a high concentration of GABA production (170.492 ppm) in MRS broth [171]. The best growing conditions that caused the highest GABA production were temperature 37 °C, pH 5.19, glutamic acid 250 mM, and time 72 h. The highest amount of GABA (195.5 ppm) after 30 days of storage was detected in whey protein drinks containing banana concentrate and stored at 25 °C.

*L. plantarum* NDC75017 (isolated from a traditional fermented dairy product from Inner Mongolia, China) was used as a starter for fermentation at 36° of Skim Milk and 80 mM L-MSG and 18 μM PLP. Under these conditions, GABA production was about 310 mg/100 g [144].

In a study conducted by Di Cagno et al., the use of *L. plantarum* DSM19463 (formerly *L. plantarum* C48) for the production of a functional grape-based beverage was evaluated [155]. The grape must, diluted with water, was enriched with yeast extract and 18.4 mM of L-glutamate and left to ferment at 30 °C. After 72 h *L. plantarum* DSM19463 synthesizes 4.83 mM of GABA [155].

In another study, the *L. plantarum* C48 has been used in sourdough fermentation [154].

The use of a blend of buckwheat, amaranth, chickpea and quinoa flours (ratio 1:1:5.3:1) subjected to sourdough fermentation by *L. plantarum* C48 allowed the manufacture of a bread enriched with GABA (504 mg/kg) [159]. The sourdough starter obtained with *L. plantarum* C48 had GABA concentrations of 12.65, 100.71 and 44.61 mg/kg for white, whole wheat and rye flours, respectively [159].

In another recent study, *L. plantarum* VL1 was used for the production of Nem Chua (traditionally Vietnamese fermented meat product). Fresh pork without fat was minced and mixed with 5% salt, 20% sugar, and 1% sodium glutamate. *L. plantarum* VL1, was added to the mixture and after 72 h of fermentation at 37 °C the meat mixture (pH 4.59) contained 1.1 mg/g of GABA [172].

In a study conducted by Nakatani et al. *L. plantarum* KB1253, isolated from Japanese pickles, is used in GABA-enriched tomato juice production [156]. This strain produces 41.0 mM GABA from 46.8 mM glutamate in tomato juice (pH 4.0, 20°Bx) incubated for 24 h at 35°.

In another study conducted by Rezaei et al., the GABA-producing strain *L. plantarum* IBRC (10817) was used in the production of a probiotic beverage made from black grapes. After 21 days, the fermented beverage had a concentration of 117.33 mg/L GABA [173].

*L. plantarum* K16 isolated from kimchi has been used to valorize some agri-food by-products [174], obtained from tomatoes, apples, oranges and green peppers. The agri-food by-products were enriched with 25 g/L of glucose, 12 g/L of yeast extract and 500 mM of MSG. Subsequently, the pH was adjusted to 5.5, and the media were inoculated with *L. plantarum* K16 and incubated at 34 °C for 96 h. *L. plantarum* K16 produced the following concentrations of GABA: 1166.81 mg/L, 1280.01 mg/L, 1626.52 mg/L and 1776.75 mg/L in apple, orange, green pepper and tomato by-products, respectively [122].

GABA is an important molecule naturally present in food matrices of plant and animal origin. However, plant-based foods contain a comparatively lower amount of GABA than animal-based foods [8,175].

Considering its potential health benefits, the studies mentioned above have shown that it is possible to increase the amount of GABA not only in some animal products but also in some fermented plant-based foods and beverages, improving their functional properties. In particular, it has been shown that through the use of *L. plantarum* as a single starter, it

has been possible to produce fermented foods from legumes, cereals, fruit juices and some agri-food by-products containing high amounts of GABA.

Besides its use as a single culture, the use of *L. plantarum* in co-culture (co-fermentation or two-stage fermentation) with other microbial strains belonging to different species is gaining increasing interest. Table 3 summarizes the relevant reports in this field.

In a study conducted by Hussin et al. [146], the effect of different carbohydrates was investigated on enhancing GABA production in yogurt cultured using a mixture of UPMC90 and UPMC91, self-cloned LAB strains (*L. plantarum* Taj-Apis362, previously isolated from the stomach of honeybee *Apis dorsata* and engineered by Tajabadi et al. [129,176]). Glucose induced more GABA production (58.56 mg/100 g) compared to inuline, FOS e GOS as prebiotics (34.19–40.51 mg/100 g), and the control sample with added PLP (48.01 mg/100 g) [146].

In other similar study, conducted by Hussin et al., self-cloned and expressed *L. plantarum* Taj-Apis362 recombinant cells, UPMC90 and UPMC91 were used to improve the GABA production in yogurt. Fermentation of skimmed milk added with glutamate (11.5 mM) after 7.25 h at 39.0 °C produced GABA-rich yogurt (29.96 mg/100 g) [177].

While many studies reported the use of single-strain LAB to generate GABA, only a few reported the production of GABA by co-culturing different bacterial strains [178].

**Table 3.** GABA production by *Lactiplantibacillus plantarum* (previously *Lactobacillus plantarum*) in co-culture with other microbial species.

| *L. plantarum* Strains | Cooperative Species/Strain | Food or Culture Medium | GABA Production | Notes | Refs. |
|---|---|---|---|---|---|
| *L. plantarum* EJ2014 | *B. subtilis* HA | pumpkin | 1.47% | Two-step fermentation | [179] |
| *L. plantarum* K154 | *B. subtilis* HA | turmeric (*Curcuma longa*)/roasted soybean meal mixture + 5% MSG | 1.78% | Two-step fermentation | [180] |
| *L. plantarum* K154 | *B. subtilis* HA | defined medium fortified with glutamate and skim milk | 4800 µg/mL | Two-step fermentation | [181] |
| *L. plantarum* K154 | *Leuconostoc mesenteroides* SM | Water dropwort | 100 mM | Two-step fermentation | [182] |
| *L. plantarum* BC114 | *S. cerevisiae* SC125 | mulberry beverage brewing | 2.42 g/L | Co-fermentation | [93] |
| *L. plantarum* GB01-21 | *C. glutamicum* G01 | cassava powder | 80.5 g/L | Two-step fermentation | [152] |
| *L. plantarum* Taj-Apis362 | *Streptococcus thermophilus* and *Lactobacillus delbrueckii* ssp. *bulgaricus* | Skim milk + 2% glucose and 11.5 mM MSG | 59.0 mg/100 g | Co-fermentation | [177] |
| *L. plantarum* K154 | *Ceriporia lacerata* | broth fortified with skim milk and 2% MSG | 15.53 mg/mL | Two-step fermentation | [127] |
| *L. plantarum* (KCTC 3105) | *Lactobacillus brevis* OPY-1 *L. acidophilus* KCCM 40265 | Soya milk | 424.67 µg/g | Co-fermentation | [157] |
| *L. plantarum* L10-11 | *Lactococcus lactis* spp. *lactis* and *Lactococcus lactis* spp. *cremonis* | milk | 11.3 mg/100 mL | Co-fermentation | [183] |
| *L. plantarum* JLSC2-6 | *Levilactobacillus brevis* YSJ3 | cauliflower stems | 35.00 mg/L | Co-fermentation | [184] |
| *L. plantarum* MCM4 | *Lactococcus lactis* subsp. *lactis* | whey-based formulate | 365.6 mg/100 mL | Co-fermentation | [185] |
| *L. plantarum* DSM749 | *L. brevis* NM101-1 | PM | 224.69 mM | Co-fermentation | [143] |
| *L. plantarum* C48 | *Lactobacillus paracasei* 15N, *Streptococcus thermophilus* DPPMAST1, *Lactobacillus delbruecki* subsp. *bulgarigus* DPPMALDb5 | Milk + 100 or mg/L of olive vegetation water phenolic extract | 67 mg/L | Co-fermentation | [186] |

Abbreviations: MSG, mono-sodic glutamate; PM, production medium (50 g/L glucose; 25 g/L peptone; 0.01 g/L MnSO$_4$. 4 H$_2$O; 2 mL Tween 80; 200 µM PLP); PLP, pyridoxal 5-phosphate.

In a study carried out by Lim et al., the co-fermentation of turmeric (*Curcuma longa*)/roasted soybean meal mixture, containing 5% MSG, was optimized to fortify it with bioactive compounds including GABA [180]. *Bacillus subtilis* HA was used for the first fermentation and *L. plantarum* K154 isolated from fermented kimchi was used for the second fermentation. The results showed that the amount of GABA increased from 0.01% before fermentation to 1.78% after the second fermentation [128].

In a further study, a two-step fermentation of pumpkin (*Cucurbita moschata*) was performed using *B. subtilis* HA and *L. plantarum* EJ2014, with the aim of producing a novel food ingredient enriched with GABA [179]. *Bacillus subtilis* HA (KCCM 10775P) strain was isolated from cheonggukjang (traditional Korean fermented soybean) while *L. plantarum* EJ2014 (KCCM 11545P) was isolated from rice bran [187]. The co-fermented pumpkin contained 1.47% GABA. *Bacillus subtilis* HA was also used in a two step-fermentation with *L. plantarum* K154, obtaining a high level of GABA production (about 4800 μg/mL) in a defined medium fortified with glutamate and skim milk [181]. Instead, Yang et al. proposed a two-step method to produce GABA from cassava powder using *C. glutamicum* G01 and *L. plantarum* GB01-21 [152]. In this study, glutamic acid was first obtained from cassava powder by saccharification and simultaneous fermentation with *C. glutamicum* G01, followed by biotransformation of glutamic acid into GABA with resting cells of *L. plantarum* GB01-21. *C. glutamicum* G01 was isolated from soil and *L. plantarum* GB01-21 was obtained through multi-mutagenesis as described in our previous study [188]. After optimizing the reaction conditions (35 °C, pH 7), the maximum concentration of GABA reached 80.5 g/L [152].

In another study, two self-cloned *L. plantarum* Taj-Apis362 strains possessing high intracellular GAD activity (UPMC90) and high extracellular GAD activity (UPMC91) and a wild-type *L. plantarum* Taj-Apis362 (UPMC1065) were co-cultured with a starter culture (a mixture of *Streptococcus thermophilus* and *Lactobacillus delbrueckii* ssp. *bulgaricus*) to produce GABA-rich yogurt [129].

The wild-type *L. plantarum* Taj-Apis362 (UPMC1065) was previously isolated from the stomach of a honeybee *Apis dorsata* [176] and used as a host for GAD gene overexpression to produce UPMC90 and UPMC91 strains. After 7 h of fermentation at 39.0 °C, the starter co-culture in skim milk with 2% glucose and 11.5 mM glutamate produces 59.00 mg/100 g of GABA.

Water dropwort (*Oenanthe javanica* DC), a common aquatic perennial plant widely cultivated in most Southeast Asian countries, was co-fermented with *Leuconostoc mesenteroides* SM and *L. plantarum* K154 to produce a novel functional food ingredient enriched with GABA (100 mM) [182]. The acidity of the fermented broth, the low concentration of sugar remaining for the second fermentation and the presence of nitrogen sources, stimulated *L. plantarum* K154 to produce GABA. These data seem to confirm that the production of GABA by bacteria is a bacterial mechanism of response towards acid stress [74].

Woraratphoka et al. used a co-culture of *L. plantarum* L10-11, *Lactococcus lactis* spp. *lactis* and *L. lactis* spp. *cremonis* in fresh cheese production [183]. *L. plantarum* L10-11 which was isolated from Thai fermented fish (Plaa-som) while *Lactococcus lactis* spp. *lactis* and *L. lactis* spp. *cremonis* they were commercial strains (Lyofast MWO030, SACCO, Italy). After 18 h the fermented milk by single-L10-11 and co-L10-11 contained 1.21 and 11.30 mg/100 mL of GABA, respectively. Thus, this suggested that in the co-culture test, by transforming lactose into lactic acid, the commercial strains decreased the pH value, creating a favorable condition for the enzymatic activity (GAD) of *L. plantarum* L10-11 that catalyzes the conversion of glutamate to GABA. Therefore, co-fermentation by *L. plantarum* L10-11 with other LAB strains could possibly increase the rate of GABA production [183].

In a previous study, it was reported that *L. plantarum* L10-11 was clearly involved in the conversion of MSG to GABA and the highest GABA production was obtained when the initial pH of MRS was in the range of 5.0–6.0 [139].

The data emerging from the above studies confirm that the optimal pH for GABA production by *L. plantarum* is placed in an acidic pH range of 4–6 [3].

Zhang et al. evaluated the effects on GABA production by co-culture of *Levilactobacillus brevis* YSJ3 and *L. plantarum* JLSC2-6. The results indicate that co-culturing these two strains can improve GABA yield (35.00 ± 1.15 mg/L) in fermented cauliflower stems (*Brassica oleracea* L. var. *botrytis*) [184].

Functional milk-based beverages enriched with 100 mg/L and 200 mg/L of olive vegetation water phenolic extract (OVWPE) were obtained via fermentation at 40 °C using *L. plantarum* C48, *L. paracasei* 15N, *S. thermophilus* DPPMAST1 and *L. delbruecki* subsp. *bulgarigus* DPPMALDb5. The highest amount of GABA (67 mg/L) was detected after 30 days at 4 °C [186].

The results obtained from the above studies have shown that co-culture fermentation using *L. plantarum* with other bacterial species is a novel technology to improve fermentation quality and promote GABA synthesis. The increase in GABA production by *L. plantarum* in co-culture with other bacteria may be related to the greater availability of nutrients released by the metabolism of the bacterium used in co-cultures [152,182] which also generates acidic end products of fermentation, which accumulate in the extracellular environment, increasing its acidity and thus promoting GABA synthesis [182–185].

Other studies, cited below, have shown that some *L. plantarum* strains improve GABA production even when used in co-culture with fungi.

Co-fermentation of *L. plantarum* K154 and fungus *Ceriporia lacerate* efficiently produced GABA (15.53 mg/mL) in a defined medium containing 3% glucose, 3% soybean flour, 0.15% MgSO$_4$, and 5% rice bran for 7 days at 25 °C [127].

The increase in GABA production in co-culture could be related to the fact that *C. lacerate*, thanks to its enzymatic activities (protease, α-amylase, cellulase, β-1,3-glucanase and phosphatase) [189], increased the availability of nutrients useful for the growth and survival of *L. plantarum*.

In a study conducted by Zhang et al., *S. cerevisiae* SC125 and *L. plantarum* BC114 were used in co-culture to ferment mulberry (*Morus alba* L.) and produce a functional beverage enriched with GABA [93]. *L. plantarum* BC114 and *S. cerevisiae* SC125 were inoculated in pasteurized mulberry substrate with 5 g/L L-glutamate and incubated at 30 °C for 72 h.

Compared to single fermentations with *L. plantarum* BC114 and *S. cerevisiae* SC125, which resulted in low GABA production (1.45 g/L and 1.03 g/L, respectively), co-culture produced a higher amount of GABA (2.42 g/L) [93].

The results obtained in this study confirm that the increased ability of *L. plantarum* to synthesize GABA could be linked to an increased availability of nutrients produced by yeasts, in particular, amino acids [190].

Therefore, co-cultures of selected fungi with GABA-producing strains belonging to *L. plantarum* species may be a promising approach for the production of GABA-enriched foods, and therefore, this biotechnological application would also merit further scientific investigation.

## 5. Conclusions and Future Perspectives

In recent decades, consumers' needs in the field of food production have increased significantly, not only in terms of organoleptic aspects but also in terms of health and well-being. Among the various functional compounds contained in foods, GABA has attracted more and more attention due to its multiple health benefits.

Although GABA is present in many foods such as fruits, vegetables and grains, its content in them is relatively low. In this context, GABA-fortified foods have been significantly considered by researchers for their important biological and functional properties. At present, GABA can be synthesized using different methods, including chemical and enzymatic synthesis, plant enrichment, and microbial production.

The numerous studies conducted on this topic highlighted that GABA production from LAB can play an important role in the food industry. In particular, fermentation by GABA-producing *L. plantarum* strains can be considered a promising possibility to increase the nutritional, sensory and functional properties of specific fermented foods.

The studies cited in this review have shown that the optimal conditions for GABA are significantly influenced by substrate composition and environmental conditions. Therefore, it is essential to optimize these parameters to improve the production of GABA, according to the production process adopted to obtain a specific fermented food.

Considering that microbial fermentation is an important technology to increase the GABA content in some fermented foods, we believe that the selection of new high-GABA-producing strains belonging to the species *L. plantarum* should remain a focus of interest in future research because it can offer concrete opportunities for the design of new functional foods.

**Author Contributions:** Conceptualization, M.I., G.P. and C.D.M.; writing—original draft preparation, M.I., G.P. and C.D.M.; writing—review and editing, M.I., G.P. and C.D.M.; visualization, C.D.M.; supervision, M.I. All authors have read and agreed to the published version of the manuscript.

**Funding:** This research received no external funding.

**Conflicts of Interest:** The authors declare no conflict of interest.

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
