# Peer review of "Biosynthesis of Gamma-Aminobutyric Acid (GABA) by Lactiplantibacillus plantarum in Fermented Food Production"

_cimb, doi:10.3390/cimb46010015_

Round 1

Reviewer 1 Report

Comments and Suggestions for Authors

General considerations

Despite not being a native English speaker, to my opinion the quality of the language (grammar and punctuation) is good, however the text doesn’t go smoothly and the transition between the paragraphs is not coherent. Additionally, the discussion of the listed facts is missing – which is inappropriate for a review article. No hypotheses are stated. The manuscript lacks also novelty because it is based only on listed observations in scientific literature.

Major issues

To my opinion, a more fluid transition between the GABA-producing bacteria and Lactiplantibacillus plantarum is needed within the abstract.

The “GRAS” status cannot be attributed to all LAB as a whole.

There are no references to the figures within the text.

Many unnecessary new paragraphs.

I would advise to discuss shortly the controversial data concerning the GABA production by Archaea.

There are no references to the tables within the text.

Minor remarks

1. Line 27: No need for a new paragraph;

2. The paragraph between lines 39-41 should be after that between lines 42-45;

3. Missing italics for the genera names in several places.

Considering the above, I do not recommend this article for publishing.

Author Response

Despite not being a native English speaker, to my opinion the quality of the language (grammar and punctuation) is good, however the text doesn’t go smoothly and the transition between the paragraphs is not coherent. Additionally, the discussion of the listed facts is missing – which is inappropriate for a review article. No hypotheses are stated. The manuscript lacks also novelty because it is based only on listed observations in scientific literature.

We thank the reviewer for the first comment. To make reading smoother, we have changed the structure of the entire manuscript, paying more attention to sequential organization and paragraph content.

In addition, we improved the discussion of the data by rewriting some sections of the manuscript and adding a more critical analysis of the cited literature. Nevertheless, we cannot agree with your comment about the lack of novelty in this revision. In fact, among the plethora of papers dealing with microbial GABA production/enrichment, this review describes, for the first time, the state-of-the-art production of GABA from L. plantarum.

Major issues

To my opinion, a more fluid transition between the GABA-producing bacteria and Lactiplantibacillus plantarum is needed within the abstract.

The abstract has been revised

The “GRAS” status cannot be attributed to all LAB as a whole.

We agree with the reviewer’ consideration and changed the sentence (L52-55)

There are no references to the figures within the text.

Indeed, references to the figures were present also in the previous version (L81 and L93 old version), in the revised version can be found in L97, L101, and L105.

Many unnecessary new paragraphs.

According to reviewer’s suggestion, the sequence of paragraphs has been changed

I would advise to discuss shortly the controversial data concerning the GABA production by Archaea.

Some paragraphs have been changed and some sentences reworded

There are no references to the tables within the text.

Although references were present also in previous version, we have added new sentences (L215-217, L317-318, and L401-403) to introduce the tables within the text.

Minor remarks

  1. Line 27: No need for a new paragraph;

The sequence of the paragraphs has been changed

  1. The paragraph between lines 39-41 should be after that between lines 42-45;

The sequence and content of some paragraphs has been changed

  1. Missing italics for the genera names in several places.

Done

Reviewer 2 Report

Comments and Suggestions for Authors

This is a very interesting manuscript dealing with the current topic of producing of gamma-aminobutyric acid (GABA) through microbial synthesis, with a particular focus on production of GABA by L. plantarum. The topic addressed in the review is original and interesting for the scientific community, and the techniques used in the work are novel.

The paper develops the subject logically. In addition, the subject of the study is of considerable interest and deserves a more in-depth dissertation.

However, some minor aspects are recommended to be analyzed by authors:

-line 137 – sufu does not need to be capitalized.

-lines 141-142 - it is recommended to assess the appropriateness of the name “wild yeast” because the microorganisms are isolated from wild flowers.

-line 151 – please analyze the appropriateness of the syntagma “red mold rice fermented”, namely if it is about red mold rice fermented or fermented red mold rice or only red mold rice?

-lines 174-177 – the microorganisms should be written in italics.

-please explain the terms / abbreviations when these are used for the first time (e.g. PLP)

-line 264 – as the authors mention that similar results have been obtained in other studies, it is recommended to add some references.

-lines 452-456 – it is not very clear if the paragraph refers to co-culture.

Author Response

This is a very interesting manuscript dealing with the current topic of producing of gamma-aminobutyric acid (GABA) through microbial synthesis, with a particular focus on production of GABA by L. plantarum. The topic addressed in the review is original and interesting for the scientific community, and the techniques used in the work are novel.

The paper develops the subject logically. In addition, the subject of the study is of considerable interest and deserves a more in-depth dissertation.

We thank the reviewer for the positive evaluation of our manuscript.

However, some minor aspects are recommended to be analyzed by authors:

-line 137 – sufu does not need to be capitalized.

Corrected

-lines 141-142 - it is recommended to assess the appropriateness of the name “wild yeast” because the microorganisms are isolated from wild flowers.

The sentence has been reworded

-line 151 – please analyze the appropriateness of the syntagma “red mold rice fermented”, namely if it is about red mold rice fermented or fermented red mold rice or only red mold rice?

The sentence has been reworded

-lines 174-177 – the microorganisms should be written in italics.

Done

-please explain the terms / abbreviations when these are used for the first time (e.g. PLP)

Done

-line 264 – as the authors mention that similar results have been obtained in other studies, it is recommended to add some references.

Done

-lines 452-456 – it is not very clear if the paragraph refers to co-culture.

The sequence and content of some paragraphs has been changed

Reviewer 3 Report

Comments and Suggestions for Authors

Dear Authors,

The present review article aimed to provide an overview on the current uses of microorganisms and in particular of L. plantarum in the production of GABA, with a detailed focus on fermented foods.

            As such, the topic of the study is interesting. However, there are major flaws in the study design, the conceptual framework and the writing of the manuscript that need to be fixed. The main concern of this manuscript relates to the “advance of thinking”. The presented review article needs to go beyond mere description and ‘state-of-the-literature’ summaries and develop new ideas and ways of thinking.

I have some improvement suggestions for Authors which are as follows:

              Please, check the spelling of different bacteria names in the whole text of manuscript.

              Why in the text of the article there is no direct connection  to the concept of “Postbiotics”?

              Have the probiotic properties of LAB bacteria a direct impact on biosynthesis of Gamma-AminoButyric Acid (GABA)?

              Lines: 280-281: why are the words bolded? Does this content have any special meaning?

              Subsection no. 8 needs to be necessarily improved that is related to the main concern – lack “advance of thinking”.

              Please thoroughly check the whole article and remove its grammatical mistakes.

              Recheck references according to the journal guidelines.

From my standpoint, this manuscript can be considered for publication in Journal – Current Issues in Molecular Biology,  after major revision,  given the above aspects.

Comments on the Quality of English Language

Minor editing of English language required

Author Response

The present review article aimed to provide an overview on the current uses of microorganisms and in particular of L. plantarum in the production of GABA, with a detailed focus on fermented foods.

            As such, the topic of the study is interesting. However, there are major flaws in the study design, the conceptual framework and the writing of the manuscript that need to be fixed. The main concern of this manuscript relates to the “advance of thinking”. The presented review article needs to go beyond mere description and ‘state-of-the-literature’ summaries and develop new ideas and ways of thinking.

We thank the reviewer for his/her comment. We improved the discussion by re-writing some sections of the manuscript and by adding a more critical analysis of the cited literature, with this latter enriched by additional references.

We have reorganized the structure of the manuscript (number and sequence of paragraphs) to make reading more fluid.

I have some improvement suggestions for Authors which are as follows:

Please, check the spelling of different bacteria names in the whole text of manuscript.

Done

Why in the text of the article there is no direct connection to the concept of “Postbiotics”?

Have the probiotic properties of LAB bacteria a direct impact on biosynthesis of Gamma-AminoButyric Acid (GABA)?

The main purpose of the review was to examine studies that took into account the production of GABA by L. plantarum at the stage of production of certain fermented foods. Therefore, the aspects related to the concept of "Postbiotics" understood aspreparation of inanimate microorganisms and/or their components that confers a health benefit on the host (Salminen, S., Collado, M.C., Endo, A. et al. The International Scientific Association of Probiotics and Prebiotics (ISAPP) consensus statement on the definition and scope of postbiotics. Nat Rev Gastroenterol Hepatol 18, 649–667 (2021). https://doi.org/10.1038/s41575-021-00440-6), it has not been addressed and discussed.

However, we believe that the concept of "Postbiotics" related to GABA production is extremely important and deserves a future and specific study. We therefore believe that the ability to produce GABA during food fermentation is to be considered a probiotic property, as evidenced in some sentences of the manuscript.

Lines: 280-281: why are the words bolded? Does this content have any special meaning?

We apologize for the bold slipped out, no special meaning. We have removed the bold.

Subsection no. 8 needs to be necessarily improved that is related to the main concern – lack “advance of thinking”.

We have amended and supplemented the paragraph

Please thoroughly check the whole article and remove its grammatical mistakes.

Done

Recheck references according to the journal guidelines.

Done

From my standpoint, this manuscript can be considered for publication in Journal – Current Issues in Molecular Biology, after major revision, given the above aspects.

Reviewer 4 Report

Comments and Suggestions for Authors

The manuscript is a review work. The topic is very interesting and current, so much so that it is valuable to publish it, but the text seems disorganized, so below I present my suggestions and advice for authors to help them improve the manuscript.

·         Abstract – In my opinion, Abstract should focus on the conclusions drawn after the literature review.

·         Sections „1. Introduction” , „2. GABA biosynthesis”, “3. GABA, an ancestral molecule”, „4. Production of GABA by Microorganism”, “5. Production of GABA by Fungi”, and „6. Production of GABA by bacteria” are an introduction to the main topic of the manuscript, so I think they should be brought together in the following sections: „1. Introduction”,  „GABA function and metabolism”, „GABA biosynthesis”. Sections „1. Introduction” and „GABA function and metabolism” are intended to explain what GABA is, what role it plays in the human body and how it is metabolized. Also worth highlighting are the potential health benefits of GABA supplementation. The “GABA biosynthesis” section should describe in detail the GABA biosynthetic pathways in bacteria, including L. plantarum. It should be clarified which enzymes are involved in this biosynthesis and which factors influence the performance of GABA synthesis. Information must be organized in a logical mental sequence.

·         Another section should address GABA production by L. plantarum. This chapter aims to present the research results on GABA production by L. plantarum under different fermentation conditions. The influence of factors such as the type of fermentation medium, temperature, pH and bacterial base on the efficiency of GABA production should be taken into account. Please treat the reference data collected critically. Make a critical but constructive analysis of the collected results.

·         Another area is the use of L. plantarum in the production of fermented foods rich in GABA. This section aims to present examples of fermented products in which L. plantarum was used to increase GABA content. You should carefully analyze and summarize the latest research on GABA biosynthesis by L. plantarum and the use of these bacteria in the production of fermented products rich in GABA. The influence of GABA content on the quality properties of these products should also be discussed.

·         Another section dealing with possible further research would be useful. This section aims to highlight areas requiring further research on GABA biosynthesis by L. plantarum and the use of these bacteria in the production of fermented GABA products. Clear and concrete conclusions from the research should be presented and directions for further research in this area suggested.

·         Section „Conclusion” – Please remember the goal of the manuscript.

·         The whole manuscript – Please note that the word "lactobacilli" is not the Latin name of the bacterium and therefore must be written with lowercase letters and without italics, while the abbreviation "LABs" means "lactic acid bacteria", so it is in the plural of “bacterium”, so must be written s “LAB”, not “LABs”.

Comments on the Quality of English Language

Minor editing of English language required. 

Author Response

The manuscript is a review work. The topic is very interesting and current, so much so that it is valuable to publish it, but the text seems disorganized, so below I present my suggestions and advice for authors to help them improve the manuscript.

We thank the reviewer for the comment. We have reorganized the structure of the manuscript (number and sequence of paragraphs) to make reading more fluid.

  • Abstract – In my opinion, Abstract should focus on the conclusions drawn after the literature review.

  • Sections „1. Introduction” , „2. GABA biosynthesis”, “3. GABA, an ancestral molecule”, „4. Production of GABA by Microorganism”, “5. Production of GABA by Fungi”, and „6. Production of GABA by bacteria” are an introduction to the main topic of the manuscript, so I think they should be brought together in the following sections: „1. Introduction”,  „GABA function and metabolism”, „GABA biosynthesis”. Sections „1. Introduction” and „GABA function and metabolism” are intended to explain what GABA is, what role it plays in the human body and how it is metabolized. Also worth highlighting are the potential health benefits of GABA supplementation. The “GABA biosynthesis” section should describe in detail the GABA biosynthetic pathways in bacteria, including L. plantarum. It should be clarified which enzymes are involved in this biosynthesis and which factors influence the performance of GABA synthesis. Information must be organized in a logical mental sequence.

According to reviewer’s suggestion, the sequence of paragraphs has been reorganized.

  • Another section should address GABA production by L. plantarum. This chapter aims to present the research results on GABA production by L. plantarum under different fermentation conditions. The influence of factors such as the type of fermentation medium, temperature, pH and bacterial base on the efficiency of GABA production should be taken into account. Please treat the reference data collected critically. Make a critical but constructive analysis of the collected results.
  • Another area is the use of L. plantarum in the production of fermented foods rich in GABA. This section aims to present examples of fermented products in which L. plantarum was used to increase GABA content. You should carefully analyze and summarize the latest research on GABA biosynthesis by L. plantarum and the use of these bacteria in the production of fermented products rich in GABA. The influence of GABA content on the quality properties of these products should also be discussed.

According to reviewer’s suggestion, the sequence of paragraphs has been rearranged and the content of some paragraphs has been supplemented with additional comments.

 Another section dealing with possible further research would be useful. This section aims to highlight areas requiring further research on GABA biosynthesis by L. plantarum and the use of these bacteria in the production of fermented GABA products. Clear and concrete conclusions from the research should be presented and directions for further research in this area suggested. Section „Conclusion” – Please remember the goal of the manuscript.

We have modified and integrated the final section entitled "Future Perspectives", emphasizing the importance of future studies on the production of GABA-rich functional foods using selected strains belonging to the L. plantarum species.

         The whole manuscript – Please note that the word "lactobacilli" is not the Latin name of the bacterium and therefore must be written with lowercase letters and without italics, while the abbreviation "LABs" means "lactic acid bacteria", so it is in the plural of “bacterium”, so must be written s “LAB”, not “LABs”.

Done

Round 2

Reviewer 1 Report

Comments and Suggestions for Authors

Again, too many unnecessary new paragraphs are present.

Line: 193: Again, NOT ALL LAB POSSESS A GRAS STATUS!

Line 204: The older name should be within the brackets.

There is now discussion or stated hypotheses in sub point 4.1.1.

Author Response

Again, too many unnecessary new paragraphs are present.

We took advantage of this suggestion and we have eliminated the 4.1.1 and 4.1.2 sub-paragraphs.

Line: 193: Again, NOT ALL LAB POSSESS A GRAS STATUS!

We perfectly agree with the reviewer and we apologize for the refuse (“the” instead of “their”). We rephrased the sentences as follows: “Because of their GRAS status, some LAB are widely used in the production of fermented foods [18] and acts as potential probiotic cultures”, We also rephrased the sentence in L189.

Line 204: The older name should be within the brackets.

We rephrased the sentences as follows: “Nowadays, Lactiplantibacillus plantarum (formerly classified as Lactobacillus plantarum) is among ….”

There is now discussion or stated hypotheses in sub point 4.1.1.

We added a comment in the revised version in L397-405.

Reviewer 3 Report

Comments and Suggestions for Authors

Dear Authors,
thank you very much for your response to my review. The revised version of the manuscript does not raise any objections. I accept it in present form.

Author Response

Dear Authors,

thank you very much for your response to my review. The revised version of the manuscript does not raise any objections. I accept it in present form.

We thank the reviewer for the positive evaluation of our manuscript.

Reviewer 4 Report

Comments and Suggestions for Authors

I see that the authors have tried to improve the manuscript according to my suggestions and I thank them for that. Unfortunately, the authors made some important substantive errors during this work, the presence of which I cannot accept in the scientific text:

·         Lines 189, 200-203  – Bifidobacteria are not LAB. They are not part of the LAB family. Please attach it to the manuscript.

·         Please do not abandon your conclusions, they are necessary for the preparation of a manuscript.

Author Response

I see that the authors have tried to improve the manuscript according to my suggestions and I thank them for that. Unfortunately, the authors made some important substantive errors during this work, the presence of which I cannot accept in the scientific text:

  • Lines 189, 200-203 – Bifidobacteria are not LAB. They are not part of the LAB family. Please attach it to the manuscript.

We perfectly agree with the reviewer and we rephrased the sentences (L189 and L200-203).

  • Please do not abandon your conclusions, they are necessary for the preparation of a manuscript.

Also in this case, we agree with the reviewer and changed the name of the last paragraph in “Conclusion and future perspectives”.